# Processing Technology and Quality Change during Storage of Fish Sausages with Textured Soy Protein

**DOI:** 10.3390/foods11223546

**Published:** 2022-11-08

**Authors:** Shuyi You, Shuqi Yang, Lanxin Li, Baodong Zheng, Yi Zhang, Hongliang Zeng

**Affiliations:** 1Engineering Research Center of Fujian-Taiwan Special Marine Food Processing and Nutrition, Ministry of Education, Fuzhou 350002, China; 2College of Food Science, Fujian Agriculture and Forestry University, Fuzhou 350002, China; 3Key Laboratory of Marine Biotechnology of Fujian Province, Institute of Oceanology, Fujian Agriculture and Forestry University, Fuzhou 350002, China

**Keywords:** textured soy protein, fish sausage, process optimization, gel strength, storage

## Abstract

The addition of textured soy protein (TSP) to surimi products extends the supply of fish protein and improves nutritional and sensory properties, which has attracted considerable research interest. In this study, a single-factor experiment and orthogonal experiment were used to determine the optimal process conditions and to assess the quality indicators of fish sausages during frozen storage. The results indicated that the optimal process conditions were as follows: the addition of 15% TSP, 8% potato starch, and 5% lard oil, resulting in a gel strength of 1894.32 g·cm. During storage of the formulation-optimized fish sausages for 180 days, the water-holding capacity, whiteness, texture properties, and gel strength of the fish sausages all decreased, whereas cooking loss, thawing loss, thiobarbituric acid reactive substances value, and total volatile base nitrogen value all increased. Consequently, TSP is beneficial to improve the gel strength and sensory score of fish sausages. The quality of fish sausages with added TSP was acceptable after storage at −18 °C, for 120 days.

## 1. Introduction

The global population is projected to exceed 9.5 billion by 2050, which would greatly increase the demand for dietary protein, potentially to an unsustainable level [1]. Fish balls, crab sticks, and fish sausages made from surimi are prized for their distinctive textures, desirable flavors, and high nutritional values [2]. The growing demand for nutritious foods that are high in protein and low in fat has greatly increased worldwide demand for surimi products [3].

World per capita seafood consumption doubled in half a century from 10 kg (live weight equivalent) in 1967 to 20 kg in 2017 [4]. Therefore, in order to alleviate the phenomenon of the over-exploitation of fish protein and obtain sufficient nutrition and ideal sensory properties, the addition of vegetable protein to surimi products has become a research hotspot [5]. The market for combined fish/vegetable protein was expected to grow at a rate of 7.9% from 2019 to 2024, with the Asia-Pacific region being the fastest growing [6]. Textured soy protein (TSP) can be used in a variety of foods, including Beyond Meat^®^ meat substitutes, sausages, ready-to-eat meals, and canned meats [7]. Materials including starch, sugar, and so on that affect the gelation and microstructure of surimi may cause a decrease in gel strength, water-holding capacity (WHC) or other gel properties [8]. The use of TSP effectively improved the amino acid score, texture, and sensory qualities of meat products [9]. Textured pea protein was substituted for meat in different proportions in low-fat frankfurter sausages, which increased their gel strength and WHC [10]. Soy protein isolates (SPI) and concentrates were widely used to enhance the WHC, textural properties, and emulsification characteristics of meat products [11]. However, there has been no report of TSP being added to surimi products.

Therefore, the formula-optimization of fish sausage with added TSP and quality changes during frozen storage were investigated in this study. Using gel strength as the dependent variable and primary quality indicator, the effects of different proportions of TSP mixture, potato starch, and lard oil on the quality of fish sausages were studied by single-factor experiments and an orthogonal experimental design.

## 2. Materials and Methods

### 2.1. Reagents and Materials

*Nemipterus virgatus* surimi, TSP, and SPI were purchased from Haixin Foods (Fuzhou, China). Sausage casings were purchased from Wuzhou Shenguan Protein Casings (Wuzhou, China). Potato starch was purchased from Zhengzhou Yuhe Food Additives (Zhengzhou, China). Lard oil was purchased from Guangzhou Meiyu Foods (Guangzhou, China). Trichloroacetic acid, magnesium oxide, boric acid solution, methyl red, methylene blue, and hydrochloric acid solution were purchased from Sinopharm Chemical Reagents (Shanghai, China). Thiobarbituric acid (TBA) solution was purchased from Nanjing Yixun Biotechnology (Nanjing, China). Frozen egg whites were purchased from Shandong Huishuo Foods (Tengzhou, China). Sugar was purchased from Shanghai Deyang Food Co., Ltd. Compound phosphate was purchased from Jinan Zhuodai Biotechnology (Jinan, China). Vegetarian meat powder was purchased from Qingdao Golden Bay Marine Biological Engineering (Qingdao, China).

### 2.2. Preparation of Fish Sausage

The fish sausages were prepared according to the method of Lago et al. [12], with some modifications. Frozen surimi was thawed overnight in a refrigerator (BCD-649WDGK, Qingdao Haier, Qingdao, China) and homogenized for 3 min in a high-speed tissue blender (JJ-2B, Daluo Scientific Instruments Co., Ltd., Shanghai, China) at 10,000 rpm. After adding 3% of the total mass of salt, blending was continued for 3 min. Then, the ingredients were added into the surimi, including sugar, egg whites, TSP mixture, and vegetarian meat powder, followed by blending for 5 min. To avoid an excessive temperature, ice water or ice cubes were added during blending. After blending, a sausage-filling machine (KL005, Kelai Catering Machinery Co., Ltd., Linyi, China) was used to fill the casings with the sausage batter to form sausages, which were then heated at 40 °C in an electric thermostatic water bath (HH-4, Changzhou Guohua Electric Appliance Co., Ltd., Changzhou, China) for 20 min. The gelatinized sausages were cooked (HB25D5L2W Steam Oven, Siemens Appliances, Shaoxing, China) at 90 °C for 30 min, then immediately cooled in ice-water and packaged with a vacuum-packing machine (DZ-350M, Shandong Mining Heavy Industry Co., Ltd., Jining, China) and stored at −18 °C.

### 2.3. Single-Factor Experiment

All percentages are expressed by weight (*w*/*w*); all sausages were made as described in Section 2.2. According to recent research by Yang [13], it was proved that when the ratio of TSP and SPI was 2:1, it could strengthen the three-dimensional network structures of fish sausages, thus the gel strength and texture properties of fish sausage were improved progressively. Three single-factor experiments were performed, varying the percentages of TSP mixture (TSP/SPI = 2:1) (5%, 10%, 15%, 20%, and 25%), potato starch (4%, 6%, 8%, 10%, and 12%), or lard oil (1%, 3%, 5%, 7%, and 9%). The gel strength of fish sausages were measured under the same conditions as other recipes.

### 2.4. Orthogonal Experimental Design

The percentages of TSP, potato starch, and lard oil added to the sausage batter were used as the three factors of the experimental design, with three levels tested to evaluate the gel strength of the resulting fish sausages, and each group of samples was tested three times in parallel. The three levels of TSP mixture, potato starch, and lard oil were determined from the single-factor experiments, and the orthogonal experimental design was L_9_(3^4^).

### 2.5. Gel Strength Measurement

The prepared fish sausages were cut into cylinders with diameters of 25 mm and heights of 30 mm. The texture analyzer (TA.XT Express, Stable Micro System, Godalming, UK) settings were: A P/5S spherical probe with a 5 mm diameter was chosen with a pre-test rate of 3.0 mm/s, a test rate of 0.5 mm/s, and a post-test rate of 3.0 mm/s; the induction force was 5.0 g, and the depression degree was 50%. The formula for calculating the gel strength was as follows:(1)$$\mathrm{Gel}\;\mathrm{strength}\;\left(g\times \mathrm{cm}\right)=\mathrm{Breaking}\;\mathrm{force}\;\left(g\right)\times \mathrm{Deformation}\left(\mathrm{cm}\right)$$

### 2.6. Sensory Evaluation

The sensory evaluation scoring standard was carried out as previously described [14], with minor modifications (Table 1). A sensory evaluation panel of 5 males and 5 females scored each group of samples on taste (25 points), smell (25 points), color (25 points), and texture (25 points). For scoring purposes, each group was counted by removing the highest and lowest scores, then calculating the average score.

### 2.7. Determination of Quality Changes during Frozen Storage

#### 2.7.1. Water-Holding Capacity

The method of Jia et al. [15] was utilized for WHC determination. Thawed sausage samples (~2 g) were accurately weighed, wrapped with filter paper (3 layers), placed in a centrifuge (Avanti J-E, Beckman Technologies, Pasadena, CA, USA), then centrifuged at 4 °C and 10,000× *g* for 10 min. The samples were removed from the tube and filter paper and weighed again. The WHC was calculated as follows:(2)$$WHC=\frac{m_{2}}{m_{1}}\times 100\%$$
where *WHC* is the water-holding capacity (%), *m*_1_ is the weight of fish sausage before centrifugation (*g*), and *m*_2_ is the weight of fish sausage after centrifugation (*g*).

#### 2.7.2. Whiteness

The method of Liu et al. [16] was utilized for whiteness determination. Each sample was cut into 5 mm thick slices, and the CIELAB color space *L**, *a**, and *b** values were measured using an automatic colorimeter (CR-10, Konica Corporation, Tokyo, Japan), standardized with standard-white and standard-black reflection plates. The whiteness value was calculated using the formula below:(3)$$W=100-[{\left({100-L}^{*}\right)}^{2}{+a}^{*2}{+b}^{*2}{]}^{1/2}$$
where *W* is the whiteness, *L** is the lightness, *a** is the redness, and *b** is the yellowness.

#### 2.7.3. Cooking Loss and Thermal Loss

Cooking loss was determined as previously described [17]. The fish sausage was cut into 2 cm slices, precisely weighed, and placed in a retort pouch. Then, it was cooked in a constant-temperature 90 °C water bath for 20 min and allowed to cool to room temperature. After removal from the retort pouch and wiping away any free water on the surface, each sample was weighed. The cooking loss was calculated using the formula below:(4)$$\mathrm{Cooking}\;\mathrm{loss}\%=\frac{\left(M_{1}-M_{2}\right)}{M_{1}}\times 100\%$$
where *M*_1_ is the mass of fish sausage before cooking (g) and *M*_2_ is the mass of fish sausage after cooking (g).

Thawing loss was determined as previously described [18]. Before thawing, the fish sausage was accurately weighed. After the fish sausage had thawed completely, the water on the surface was wiped away with filter paper before weighing again. The thawing loss was calculated as follows:(5)$$\mathrm{Thawing}\;\mathrm{loss}\%=\frac{\left(M_{3}-M_{4}\right)}{M_{3}}\times 100\%$$
where *M*_3_ is the mass of the fish sausage before thawing (g) and *M*_4_ is the mass of the fish sausage after thawing (g).

### 2.8. Evaluation of Texture Properties

Texture properties was determined as previously described [19]. The prepared fish sausages were left to equilibrate at room temperature for 1 h, then cut into 25 mm diameter × 30 mm cylinders. The hardness, springiness, cohesiveness, and chewiness of the fish sausage samples were measured with the TA.XT Express texture analyzer with the following settings: P/36R probe; test mode TPA; pre-test rate 3.0 mm/s; test rate 0.5 mm/s; post-test rate 3.0 mm/s; induction force 5.0 g; depression degree 50%.

### 2.9. Evaluation of TBARS Value and TVB-N Value

#### 2.9.1. TBARS Value

The method of Siripattrawan et al. [14] was utilized for TBARS determination. Frozen fish sausage samples (5 g) with a trichloroacetic acid solution (25 mL, 20% *w*/*v*) and distilled water (20 mL) were homogenized for 1 min, left to stand at room temperature, then centrifuged at 6000× *g* for 10 min. The supernatant was retained and made up to 50 mL with distilled water, then an aliquot (10 mL) was mixed with TBA solution (10 mL, 0.02 mol/L), heated in a boiling water bath for 40 min, cooled, and the absorbance at 532 nm determined (UV-1780 spectrophotometer, Suzhou Daojin Instrument Co., Ltd., Suzhou, China). The TBARS value was calculated as follows:(6)TBA=A×7.8
where *A* is the absorbance of the sample at 532 nm, 7.8 is the conversion factor, and the unit of TBARS value is mg MDA/kg.

#### 2.9.2. TVB-N Value

The method of FAN et al. [20] was utilized to obtain the TVB-N value, with some modifications. The fish sausage (10 g) was mixed with distilled water (100 mL) for 30 min, then filtered. The filtrate (5 mL) was mixed with aqueous magnesium oxide (1% *w*/*v*, 5 mL) and boiled for 5 min. The distillate was absorbed by 20 g/L boric acid, and then the boric acid solution was titrated by calibrated 0.01 M HCl with methyl red and methylene blue as mixed indicators. The calculation was as follows:(7)$$TVB-N\;\mathrm{mg}/100\;g=\frac{(V_{1}-V_{2})\times C\times 14}{m\times 5/100}\times 100$$
where *V*_1_ is the volume of the HCl used for the titration of sample (mL), *V*_2_ is volume of the HCl for the titration of blank (mL), *C* is the concentration of HCl (mol·L^−1^), and *m* is the weight of fish sausage (g).

### 2.10. Statistical Analysis

All experimental measurements were conducted at least in triplicate. The Statistical Package for Social Science 26.0 software (SPSS Inc., Chicago, IL, USA) was used to analyze the experimental data, and the mean ± standard deviation (SD) was provided. Duncan’s multiple range test method was used to compare significant differences using one-way analysis of variance (ANOVA) at the 0.05 level. Charts were prepared using Origin 9.0 (OriginLab Corporation, Northampton, MA, USA).

## 3. Results and Discussion

### 3.1. Results of Single-Factor Experiments

#### 3.1.1. Effect of TSP Mixture Addition on Gel Strength of Fish Sausages

The quality of surimi product, consumer acceptance, and economic value are strongly influenced by its gel properties and flavor [21]. The gel network structure formed by protein denaturation and aggregation determines the gel strength of surimi gels [22].

When the addition of potato starch and lard oil were fixed at 8% and 5%, the sensory score was low at both 5% and 25% (76.38 and 77.5, respectively), and peaked at 15% with a score of 91.63, which was significantly higher (*p* < 0.05, Figure 1A-a. According to the findings of Liu et al. [23], soy protein could significantly improve the gel strength and flavor. It appeared that too little TSP mixture could weaken the gelling and emulsifying properties of the surimi, whereas too much TSP mixture could impart a strong smell of soybeans, masking the natural flavor of the surimi and reducing the sensory quality. According to the findings of Pardi et al. [24], the use of soy protein should be carefully controlled, because an excessive addition produced bitter-tasting coumaric and ferulic acid derivatives in the product during heat treatment, which impaired the overall taste.

Gel strength, another important indicator of surimi quality, which was low at both 5% and 25% (822.45 g·cm and 996.10 g·cm, respectively), and peaked at 15% with a value of 1926.11 g·cm, which was significantly higher (*p* < 0.05, Figure 1B-a. It appeared that adding the optimal proportion of TSP during the heating and gelling process can markedly increase the density of the surimi gel network and make the tissue firmer. When the TSP:SPI ratio was 2:1, the proteins efficiently filled the tissue structure of the fish sausage and generated a stable and dense composite gel system. When too little TSP mixture was added, the interaction with myosin was insufficient, and the gel network structure was not compact and stable. This was in accordance with the study by Lempek [25], in which the gel strength of surimi decreased with the increased addition of soy protein, because competition for water inhibited the gelling of the surimi protein and weakened the resulting gel.

#### 3.1.2. Effect of Potato Starch Addition on Gel Strength of Fish Sausage

When the addition of TSP and lard oil were fixed at 15% and 5%, the sensory evaluation score of the fish sausages varied with the proportion of potato starch added, similarly to TSP mixture, i.e., low scores at 4% and 12% (77.63 and 82.03, respectively) and peaking at 8%, with a score of 93.75 (Figure 1A-b. This agreed with the findings of RAHMAN et al. [26], in which an addition of 8% potato starch appeared to maximize its own gelling ability, thereby improving the elasticity and texture of the fish sausage. However, if too much was added, the fish sausage was too stiff, reducing its sensory score.

The gel strength, breaking force, and deformation of fish sausages increased initially and then decreased as the content of soybean protein mixture increased, and all peaked at 8% potato starch addition, with values of 1812.14 g·cm, 1677.90 g, and 1.08 cm, respectively (Figure 1B-b. The gel strength with 5% and 25% potato starch decreased significantly (*p* < 0.05) to 913.45 g·cm and 976.10 g·cm, respectively. According to the findings of Guo et al. [27], which founded that an appropriate proportion of potato starch absorbed water and expanded during heating, forming a three-dimensional network structure that strengthened the surimi gel. However, excessive potato starch could cause proteins and starch to compete for moisture, leading to inadequate starch swelling and to the dehydration of the surimi protein.

#### 3.1.3. Effect of Lard Oil Addition on Gel Strength of Fish Sausages

When the addition of TSP and potato starch were fixed at 15% and 8%, respectively, the sensory score varied with the proportion of lard oil added, similarly to TSP mixture, i.e., low scores at 1% and 9%, peaking at 5% with a score of 93 (Figure 1A-c. It appears that lard oil can effectively retain the desirable flavor compounds in the surimi, while inhibiting its undesirable fishy smell to some extent. With 1 and 9% lard oil, the sensory scores decreased significantly (*p* < 0.05) to 71.63 and 77.38, respectively. It appeared that insufficient lard oil inadequately emulsified the surimi protein, TSP, and starch, resulting in a dry texture, but excellent taste and a lower overall sensory score. Too much lard oil resulted in cavities in the sausage filling and aggregations of fat, giving an uneven texture and a low sensory quality.

The gel strength, breaking force, and deformation of fish sausage all peaked at 5% lard oil addition, with values of 1801.21 g·cm, 1652.49 g, and 1.09 cm, respectively (Figure 1B-c. It appeared that lard oil functions as an emulsifier, interacting with protein molecules and moisture to enhance the textural smoothness, aroma, and taste of fish sausages. The gel strength decreased significantly (*p* < 0.05) with 1% and 9% lard oil to 1086.40 g·cm and 1164.80 g·cm, respectively. The emulsification effect was insufficient when too little lard oil was added, so it could not effectively interact with protein molecules and water, lowering the gel strength. Too much lard oil decreased the hardness of the fish sausage and weakened the network structure and gel strength.

### 3.2. Analysis of Orthogonal Experiment Results

Since the trend of sensory score and gel strength peaked at the same proportions of the three additives in the single-factor experiments, gel strength was used as the representative measurement index for the experimental design tests. Based on the single-factor results (Section 3.1), the central values of the TSP mixture (A), potato starch (B), and lard oil (C) in fish sausage were set at 15%, 8%, and 5%, respectively. On this basis, the orthogonal test design was an L_9_(3^4^) table with the gel strength as the measurement index, as shown in Table 2.

The experimental results are shown in Table 3; the R values indicate that the order of influence on the gel strength of fish sausage was: addition of TSP mixture (A) > addition of potato starch (B) > addition of lard oil (C). The optimal ingredient formulation was A_2_B_2_C_2_, yielding an optimum gel strength of 1894.32 g·cm.

### 3.3. Fish Sausage Quality Changes during Frozen Storage

#### 3.3.1. Water-Holding Capacity

Figure 2A conveys the change in WHC during fish sausage storage, which shows a decreasing trend. The WHC decreased during fish sausage storage, from 93.68% initially to 76.77% after 120 days (a significant decrease (*p* < 0.05)), then did not change significantly (*p* > 0.05) for up to 180 days. It may because once the TSP:SPI ratio was appropriate, the TSP may have increased the covalent cross-linking between the surimi and the soy proteins, improving the ability of the gel structure to retain water. In addition, an appropriate amount of starch can passivate the free water in the fish sausage and make it difficult to precipitate, thereby increasing the WHC. The space left by melted ice crystals remained in the gel because water separated by freezing cannot be completely absorbed by the sausage filling after thawing [28]. Microorganisms had a considerable impact on the internal structure of fish sausages in the first 4 months, decomposing most of the nutrients such as amino acids and fats, resulting in the looseness reaching its maximum value, most likely due to the problem of microbial contamination during the processing and storage of fish sausage.

#### 3.3.2. Whiteness

Whiteness is a major indicator of fish sausage quality [29], and it has a significant impact on consumer acceptability. Figure 2B depicts the change in whiteness of fish sausages during storage. There was no significant difference in whiteness after 7 days of storage (*p* > 0.05); at 120 days, it was significantly lower (*p* < 0.05) at 43.88, compared with 54.00 initially, then did not change significantly for up to 180 days (*p* > 0.05). It could be because the color of TSP was white, especially after rehydration, and appeared similar to surimi, hence it had a slight effect on the whiteness of the fish sausages. Luo et al. [30] reported that the expansion of large-sized starch granules during the heating process resulted in the decrease of whiteness. The loss of whiteness during storage appears to result from fat oxidation by the addition of lard oil which would have increased the yellowness value and possibly from the Maillard reaction and secondary oxidation.

#### 3.3.3. Cooking Loss and Thawing Loss

The extent of drip loss indicates the strength of the gel network; the higher the crosslinking of the gel, the lower the drip loss [31]. Figure 2C depicts the effect of fish sausages on cooking loss at various storage times, which shows that the cooking loss for fresh fish sausage was 25.48%. This was consistent with the findings of Gujral et al. [32], and it demonstrated that TSP improved the cohesion and adhesion of the patties, reducing cooking losses and preventing excessive shrinkage. After 120 days, the cooking loss of fish sausages was 38.03%, significantly higher than that of fresh fish sausage (*p* < 0.05), and this did not change significantly for up to 180 days (*p* > 0.05). The heating of the thawed fish sausages may have degraded the protein network structure, thus destabilizing the gel and increasing the water loss.

The variation of thawing loss with storage time was almost identical to that of cooking loss in Figure 2D. The thawing loss of fish sausage was significantly lower than that of fresh fish sausages when stored for 7–120 days (*p* < 0.05). At 180 days, the thawing loss was not significantly different from that of the fish sausage stored for 150 days (*p* > 0.05). Although exogenous additives such as TSP and potato starch maximized water retention during the preparation of the fish sausages, structural changes during storage could “loosen” the gel structure, resulting in lower water retention. As according to Vieira et al. [33], both frozen storage and thawing reduced the WHC of fresh meat and meat products, resulting in thawing losses.

According to the above findings, the storage time of fish sausages should not exceed 120 days, as this will result in a decrease in product quality and nutritional level.

### 3.4. Changes in the Texture Properties of Fish Sausage during Frozen Storage

Long-term frozen storage can lead to changes in the physicochemical properties and structures of surimi [34]. Figure 3 depicts the results, which show a decreasing trend. Fresh fish sausage had the highest hardness, elasticity, cohesion, and chewiness, with values of 7340.20 g, 0.93, 0.62, and 4199.57, respectively. After 7 days, there was no significant difference in any index when compared to fresh fish sausage (*p* > 0.05). It demonstrated that the fully denatured TSP combines well with the surimi protein, forming a compact, robust network structure. Meanwhile, when combined with surimi and lard oil, the appropriate amount of soybean protein fully utilized its emulsifying capacity, which promoted the synthesis of protein molecules into gel systems and improved the TPA parameters. All four textural indices decreased significantly (*p* < 0.05) between 7 and 120 days. This agreed with a previous report in which the TPA parameters of fish sausages were affected by storage time at 10 °C, with hardness, chewiness, and elasticity all decreasing significantly as the storage time increased [12]. This might be the case because prolonged low-temperature freezing altered the degree to which the protein content of *Nemipterus virgatus* and soy protein mixture was cross-linked during the gelation process, leading to overly wide pores inside the fish sausage structure, an unstable gel network structure, and poor elasticity. In addition, with the loss of moisture during storage, the myofibrillar proteins of surimi will be diluted by an excessive concentration of potato starch, which could then cause phase inversion and a loss of textural qualities [30]. The springiness of the fish sausages stored for 180 days was not significantly different from that of the fish sausages stored for 150 days (*p* > 0.05), presumably because structural degradation had reached a steady state by 150 days. It has been shown that when fish sausages were stored for 30–120 days, the texture properties were greatly reduced.

### 3.5. Changes in Gel Strength during Frozen Storage of Fish Sausage

The quality of surimi product and economic value are heavily influenced by its gel strength [21]. Figure 4 depicts the effect of different storage times on the gel strength of fish sausages. The gel strength, breaking force, and deformation of fresh fish sausages were 1894.32 g·cm, 1685.08 g, and 1.12 cm, respectively. After 90 days of storage, the gel strength of fish sausages significantly decreased to 986.64 g·cm (*p* < 0.05). Surasani et al. [35] reported that the expressible moisture content of meat sausages increased with freezing time, resulting in a decrease in gel strength which was more pronounced in sausages prepared without added SPI. The gel strength after 120–180 days was not significantly different from that after 90 days (*p* > 0.05), presumably because structural degradation had stopped by 90 days. Proteins in fish sausage oxidize and degrade during frozen storage as the hydratability of the proteins decreased, disrupting the gel network [36]. As a result, the fish sausages should be consumed within 4 months of purchase, as long-term storage will result in a significant decrease in gel strength and quality.

### 3.6. Changes in TBARS Values and TVB-N Values of Fish Sausages during Storage

#### 3.6.1. TBARS Value

The TBARS value is used to assess the degree of oxidative rancidity of fish sausages based on the pink product of its reagent’s interaction with malondialdehyde [37]. The effect of the fish sausages on the TBARS values over time is depicted in Figure 5A, with an increasing trend. Fresh fish sausages had the lowest TBARS value at 0.092 mg MDA/kg, then the value significantly increased to 0.31 mg MAD/kg after 120 days (*p* < 0.05). The significant amount of saturated fatty acids present in the lard oil added to the fish sausages was directly related to the drop in the TBARS value. As was consistent with the findings of Surasani et al. [35], the TBARS values of meat sausages packaged in cellulose and high pressure-low density polyethylene increased from 0.18 to 0.32 mg MAD/kg during three months of frozen storage when formulated with SPI. However, this was not significantly different after 150 days of storage (*p* > 0.05). During the heating process of the fish sausages, TSP and surimi protein formed a matrix encapsulating fat, which was possibly conducive to the preservation of the fish sausages. It appeared that long-term freezing exposed the unsaturated fatty acids to oxygen, forming fatty acid hydroperoxides and their breakdown products; however, oxidation was essentially complete after 120 days.

#### 3.6.2. TVB-N Value

The increase in the TVB-N values in fish is thought to be due to bacterial metabolism and the oxidation of protein and non-protein components [20]. Figure 5 depicts the effect of fish sausages on TVB-N value at various storage times (A). Fresh fish sausage had the lowest TVB-N value at 7.08 mg/100 g, which then significantly increased to 29.52 mg/100 g after 120 days of frozen storage in an exponential manner (*p* < 0.05), but which then did not significantly increase further (*p* > 0.05). This could be mainly because microorganisms may multiply during the freezing process, degrading the amino acids in the fish protein after microbial contamination during sausage processing. Maheshwara et al. [38] reported that the TVB-N value of cold-stored fish sausages decreased as the storage time increased. As a result, the best cosumption time of fish sausages was determined by a comprehensive evaluation of frozen storage time within 120 days. 

## 4. Conclusions

The optimal production process of TSP fish sausages and their storage quality changes were experimentally investigated in this paper. It was concluded that the optimum technological conditions of fish sausages were: 15% TSP mixture, 8% potato starch, and 5% lard oil. Under these conditions, the gel strength of fish sausage was 1894.32 g·cm. Moreover, during frozen storage for 180 days, the WHC, whiteness, textural properties, and gel strength of the fish sausages all decreased, whereas the cooking loss, thawing loss, TBARS value, and TVB-N value all increased. Consequently, the quality of the fish sausages with TSP remained relatively good after 120 days at −18 °C. This paper provides a theoretical foundation for the further optimizing of fish sausage formulation and processing and for minimizing the quality changes during storage.

## Figures and Tables

**Figure 1 foods-11-03546-f001:**
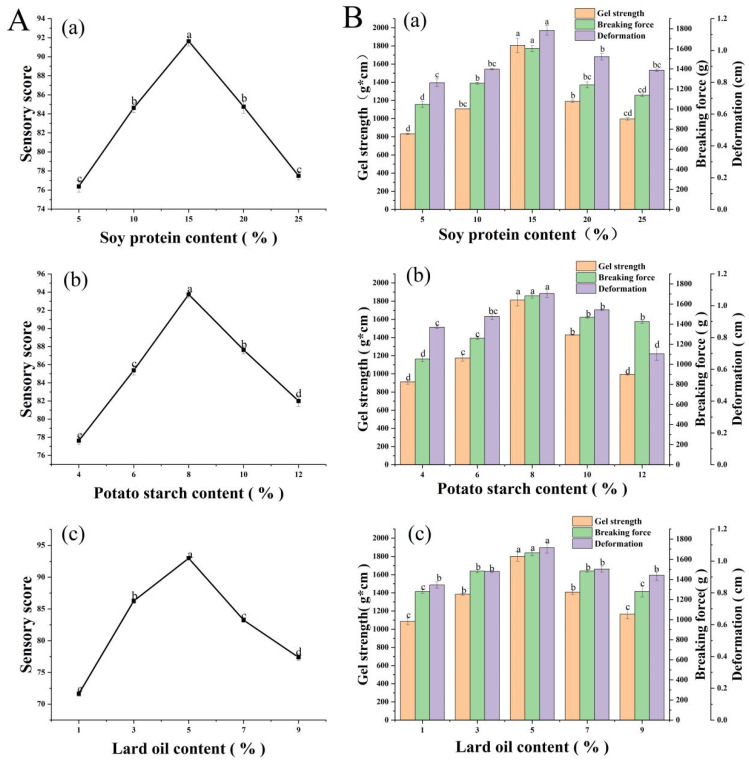
Changes in sensory score (**A**) and gel strength (**B**) of fish sausages with different textured soy protein (TSP) mixture content (**a**), potato starch content (**b**), and lard oil content (**c**). Significant differences between samples *(p* < 0.05) are denoted by lowercase letters.

**Figure 2 foods-11-03546-f002:**
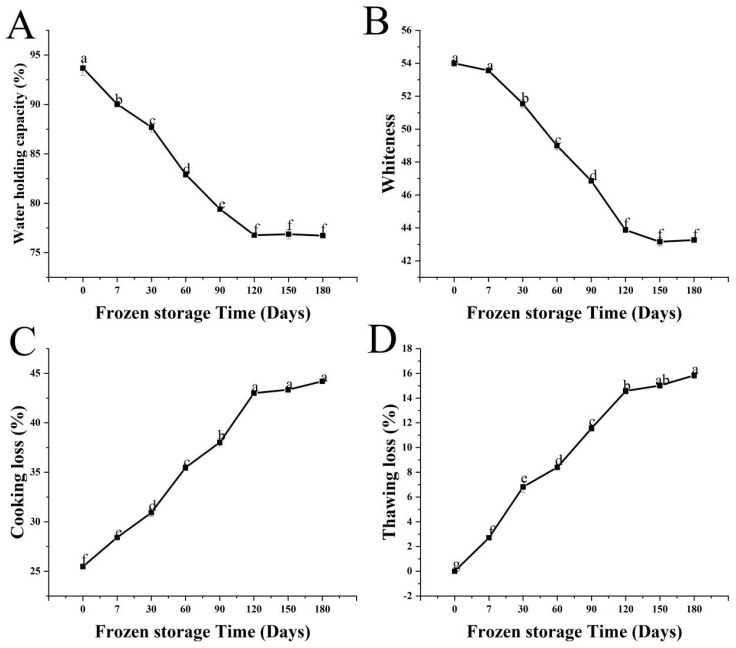
Changes in the water-holding capacity (**A**), whiteness (**B**), cooking loss (**C**), and thawing loss (**D**) of fish sausages during frozen storage. Significant differences between samples (*p* < 0.05) are denoted by lowercase letters.

**Figure 3 foods-11-03546-f003:**
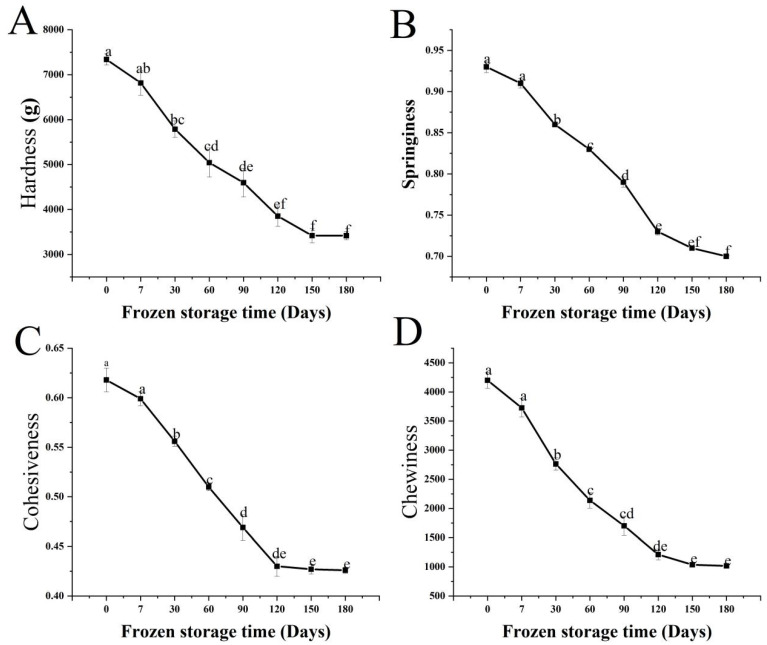
Changes in the hardness (**A**), springiness (**B**), cohesiveness (**C**), and chewiness (**D**) of fish sausages during frozen storage. Significant differences between samples (*p* < 0.05) are denoted by lowercase letters.

**Figure 4 foods-11-03546-f004:**
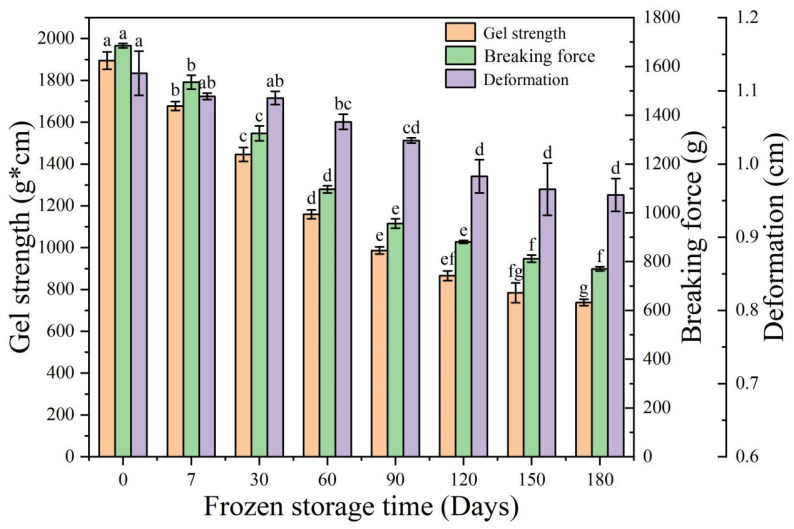
Changes in the gel strength, breaking force, and deformation of fish sausages during frozen storage. Significant differences between samples (*p* < 0.05) are denoted by lowercase letters.

**Figure 5 foods-11-03546-f005:**
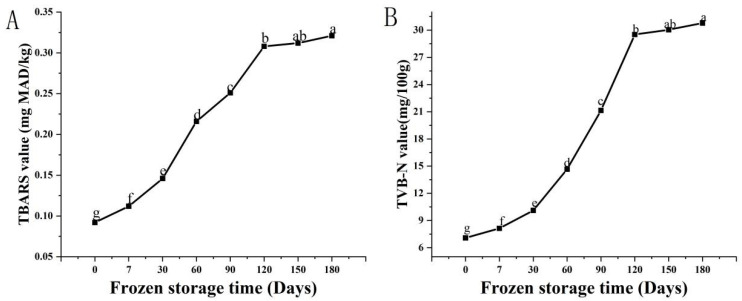
Changes in Thiobarbituric acid reactive substances (TBARS) value (**A**) and total volatile base nitrogen (TVB-N) value (**B**) of fish sausages during frozen storage. Significant differences between samples (*p* < 0.05) are denoted by lowercase letters.

**Table 1 foods-11-03546-t001:** Sensory score standards.

Test Items	Score Standards	Sensory Score
Taste (25)	Gritty, foreign body sensation in chewing, poor taste, too salty or too bland.	0~8
	Slightly gritty chewing, no foreign body sensation, moderate saltiness.	9~17
	Chewing without sandy feeling, no foreign body feeling, just right saltiness.	18~25
Smell (25)	Bad smell.	0~8
	Slight beany and fishy smell.	9~17
	Mixed fragrance.	18~25
Color and lustre (25)	Dark color, yellow-gray, matt.	0~8
	Medium bright color, yellow–white, slightly glossy.	9~17
	Brighter color, yellow–white, shiny.	18~25
Texture (25)	The surface is normal smooth, the structure is loose with many holes, the TSP is not evenly mixed, and no elasticity.	0~8
	The surface is smooth, the structure is loose with individual holes, the blending of the TSP is unevenly mixed, and the elasticity is poor.	9~17
	The surface is smooth, the structure is dense with no pores, the TSP is evenly mixed, and the elasticity is good.	18~25

**Table 2 foods-11-03546-t002:** Orthogonal test factor level table.

Level	Factor		
	A (Textured Soy Protein Mixture)/%	B (Potato Starch)/%	C (Lard Oil)/%
1	10	6	3
2	15	8	5
3	20	10	7

**Table 3 foods-11-03546-t003:** Orthogonal experimental design results.

Number	A (Textured Soy Protein Mixture)/%	B (Potato Starch)/%	C (Lard Oil)/%	Null Columns	Gel Strength (g*cm)
1	1	1	1	1	1789.86
2	1	2	2	2	1860.99
3	1	3	3	3	1811.49
4	2	1	2	3	1848.29
5	2	2	3	1	1892.37
6	2	3	1	2	1817.55
7	3	1	3	2	1691.65
8	3	2	1	3	1824.22
9	3	3	2	1	1741.92
K_1_	5462.34	5329.80	5431.63		
K_2_	5558.21	5577.58	5451.20		
K_3_	5257.79	5370.96	5395.51		
The optimal level	A_2_	B_2_	C_2_		
R	300.42	247.78	55.69		

## Data Availability

Data is contained within the article.

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
