# Peer review of "Processing Technology and Quality Change during Storage of Fish Sausages with Textured Soy Protein"

_foods, 2022, doi:10.3390/foods11223546_

Round 1

Reviewer 1 Report

The authors proposed an application of TSP to surimi products to improve food quality. The theme fits the journal scope well and may contribute to the development of food technology, in my opinion. The authors designed a set of optimizing experiments, found the best conditions, and then validated the effects. The experimental part looks reasonable, and the results seem sound. But the discussion part looks insufficient: such findings were not explained in depth. There are also some issues that the authors should pay attention to:

1. It is known that the cold storage of surimi products may cause food quality problems. And the specific issues are related to the chemical characteristics of the surimi products, which means using different food materials may lead to different findings. Have the authors considered this point? Even if it is not concluded in the experiment design, the authors should mention it in the discussion part.

2. What made TSP effective for surimi product preservation? It must be carefully discussed. Accordingly, the chemical profile of surimi, TSP, and others should be provided as introductory and necessary information.

3. The authors tested the TBARS value. Does it mean that the authors consider TSP as an antioxidant against cold storage-induced oxidation? Please explain this idea in more detail.

4. Is this beneficial effect dose-dependent? Or, please propose a strategy (not specific formula or recipe) for this theme to improve the scientific depth of this work.

5. Language & writing: there are not a few format issues, grammar errors, and spelling mistakes.

Author Response

Dear reviewer,

A revision on foods-1955244 has been carried out. Replies to the reviewer 1’ comments are listed below and the corresponding corrections were made in the revised manuscript and uploaded an attachment. We have marked all the changes in red in the revised manuscript.

Q1. It is known that the cold storage of surimi products may cause food quality problems. And the specific issues are related to the chemical characteristics of the surimi products, which means using different food materials may lead to different findings. Have the authors considered this point? Even if it is not concluded in the experiment design, the authors should mention it in the discussion part.

Answer 1: Comment has been taken into account. Indeed, the use of different food materials will affect the quality of fish sausage. The main focus of this article was not the differences in the storage process of fish sausages with different methods, different parameters, and different component ratios; instead, it primarily focused on the quality changes of fish sausages produced under the ideal formula. However, the processing quality of various components of fish sausage is ready for publication in the next article. We have added the relevant information in the discussion part according to your suggestions (Lines 219-221, 228-230, 266-268, 281-285, 300-302, 320-326, 357-359).

Q2. What made TSP effective for surimi product preservation? It must be carefully discussed. Accordingly, the chemical profile of surimi, TSP, and others should be provided as introductory and necessary information.

Answer 2: Comment has been taken into account. In fact, other ingredients, not just TSP, are responsible for changes in the chemical profile of fish sausages during storage. This paper focused on the optimal formula of fish sausage and its quality changes during storage were obtained by using TSP and other fixed factors. Moreover, during the heating process of fish sausages, TSP and surimi protein formed a matrix encapsulating fat, which possibly conducive to the preservation of fish sausages. We have added the relevant information in the discussion part according to your suggestions (Lines 265-268, 270-274, 281-285, 300-302, 320-326, 344-346, 357-359, 363-364).

Q3. The authors tested the TBARS value. Does it mean that the authors consider TSP as an antioxidant against cold storage-induced oxidation? Please explain this idea in more detail.

Answer 3: Comment has been taken into account. When developing quality parameters for all surimi, TBARS will be taken into consideration. In this paper, gel strength, texture properties, TVB-N and so on were also measured to evaluate the quality of fish sausage. TSP is not regarded in this article as an antioxidant against oxidation brought on by cold storage. TSP addition is only tangentially connected to TBARS, which is mostly related to fat oxidation. TSP inhibits fat oxidation-TBARS and encapsulates oil. But whether TSP has antioxidant effect remains to be studied.We have added the relevant information in the discussion part according to your suggestions (Lines 363-364).

Q4. Is this beneficial effect dose-dependent? Or, please propose a strategy (not specific formula or recipe) for this theme to improve the scientific depth of this work.

Answer 4: Comment has been taken into account. The best dose ratio of TSP and soy protein isolate added to fish sausages has been studied in the previous period, and the properties have been determined. It is prepared to be posted in another article due to the substantial material. According to the dosage ratio of the previous study, this paper mainly studied the quality change of fish sausage under the selected optimal dosage. Moreover, it’s consistent with the industrial study of changes in the quality of a food product.

Q5. Language & writing: there are not a few format issues, grammar errors, and spelling mistakes.

Answer 5: Comment has been taken into account. They were revised accordingly (Lines 43-44,71-73, 133-134, 219-221, 247-249,265-266, 320-323, 374-376). Our manuscript has been reviewed by a native English speaker and revised to improve readability.

To sum up, we have seriously considered your questions, and made modifications and explanations one by one, and the article has been improved, please do not hesitate to contact us if there are any question. Thanks again to the reviewers and editors for your hard work! Best wishes to you!

Address: Fujian Agriculture and Forestry University, College of Food Science, Fuzhou, Fujian, P. R. China 350002.

Tel.: +86 591 83736738

fax: +86 591 83739118

E-mail address: [email protected] (Y. Zhang), [email protected] (H. Zeng)

Sincerely,

Zhang Yi and Hongliang Zeng

17-Oct-2022 

Reviewer 2 Report

1. The fine-tuning of English is very important in the entire document 

2. The abstract should be revised, not mentioning the complete study

3. What is this g.mm unit? recheck once

4. Line 24, missing degree symbol 

5. Line 35 can get the data of the fish meat instead of the animal meat?

6. Line 43, what are amino acid scores?

7. The sentence in lines 43 and 44 should be changed 

8. Lines 51 to 55 should be removed it is a methodology, not an introduction

9. Still introduction should be elaborate on previous studies and the effect of the composition 

10. What is the variety of the Surimi, collected? How much was collected? what is the sample prepared amount? How much is stored?

11.  Section 2.2. These are your developed methods or taken from some reported studies? give the citation 

12. Line 68, what amount of salt was added?

13. Line 69, what are the ingredients?

14. Line 75, where you packed, and how you packed? explain clearly. Mention the details here. 

15. what is your single factor here? It looks like it is multifactor. 

16.  Line 101, They are trained or untrained? These 10 panelists are enough? 

17. Give citations for the 2.7.1

18. Provide the citation for section 2.7.2

19. Line 123, which bag and how?

20. Provide the citation for section 2.7.3

21. Section 2.8 is how what is different from section 2.5.

22.  Provide the citation for section 2.8

23. What is section 3 on page 5 that is given only results? usually, it should be  Results and the discussion or discussion section should be there, but I can not find it. 

24. In section 3.1.1. given the sensory and gel strength together?

25. Line 190, Citation is given in lowercase

26. Line 193, what is the sensory score? why not given? 

27.  Discussion is very poor need to work on the discussion of the results.

28.  conclusions should be fine-tuned.  

29. The references should be revised properly. for instance check the 33 number the title of the paper is upper case.

Author Response

Dear reviewer,

A revision on foods-1955244 has been carried out. Replies to the reviewer 2’ comments are listed below and the corresponding corrections were made in the revised manuscript and uploaded an attachment. We have marked all the changes in red in the revised manuscript.

Q1. The fine-tuning of English is very important in the entire document 

Answer: Comment has been taken into account. They were revised accordingly  (Lines 43-44,71-73, 133-134, 219-221, 247-249,265-266, 320-323, 374-376). The grammar in the whole text has been checked and revised by a native English expert.

Q2. The abstract should be revised, not mentioning the complete study

Answer: Comment has been taken into account. It was revised accordingly (Lines 12-23).

Q3. What is this g.mm unit? recheck once

Answer: Comment has been taken into account. Internationally,  ‘g·mm’ has considered as the universal unit of gel strength. According to some references,they used ‘g·mm’ as the unit of gel strength. References are as follows:

  1. Yi, S.; Huo, Y.; Qiao, C.; Wang, W.; Li, X. Synergistic gelation effects in surimi mixtures composed of Nemipterus virgatus and hypophthalmichtys molitrix. Journal of Food Science2019, 84, 3634-3641, doi:10.1111/1750-3841.14761.
  2. Ma, X.-S.; Yi, S.-M.; Yu, Y.-M.; Li, J.-R.; Chen, J.-R. Changes in gel properties and water properties of Nemipterus virgatussurimi gel induced by high-pressure processing. LWT - Food Science and Technology 2015, 61, 377-384, doi:10.1016/j.lwt.2014.12.041.

Q4. Line 24, missing degree symbol 

Answer: Comment has been taken into account. In the revised manuscript, -18 C was changed to -18 °C (Line 23).

Q5. Line 35 can get the data of the fish meat instead of the animal meat?

Answer: Comment has been taken into account. It was supplymented accordingly (Lines 33-34, 37-39).

Q6. Line 43, what are amino acid scores?

Answer: Comment has been taken into account. Amino acid score (AAS), also known as protein chemistry score, is a widely used method for evaluating the nutritional value of food protein.

Q7. The sentence in lines 43 and 44 should be changed 

Answer: Comment has been taken into account. We have revised the manuscript as suggested (Lines 43-44).

Q8. Lines 51 to 55 should be removed it is a methodology, not an introduction

Answer:Comments have been taken into account. They were removed accordingly.

Q9. Still introduction should be elaborate on previous studies and the effect of the composition.

Answer: Comments have been taken into account. They were supplemented accordingly (Lines 39-41, 46-48).

Q10. What is the variety of the Surimi, collected? How much was collected? what is the sample prepared amount? How much is stored?

Answer: Comments have been taken into account.The surimi we used was Nemipterus virgatus surimi, and the sample preparation amount is 20 kg, 22 CNY per kilo.

Q11.  Section 2.2. These are your developed methods or taken from some reported studies? give the citation 

Answer: Comments have been taken into account. The methods of making fish sausages was taken from reported studies. It was supplemented accordingly (Line 69).

Q12. Line 68, what amount of salt was added?

Answer: Comments have been taken into account. It was supplemented accordingly (Lines 72-73). The amount of salt added was 3% of the total mass.

Q13. Line 69, what are the ingredients?

Answer: Comments have been taken into account. They were supplemented accordingly (Lines 64-67, 73-74).

Q14. Line 75, where you packed, and how you packed? explain clearly. Mention the details here. 

Answer: Comments have been taken into account. It was supplemented accordingly (Lines 81-82). The gelatinized sausages were cooked at 90 °C for 30 min, then immediately cooled in ice-water and packaged with a vacuum packing machine , stored at -18 °C.

Q15. what is your single factor here? It looks like it is multifactor. 

Answer: Comments have been taken into account. Single factor experiment refers to the optimal process conditions of fish sausage through different TSP, potato starch and lard oil additions. For example, when a single factor experiment was conducted on the amount of TSP added, other factors were fixed. They were supplemented accordingly in discussion part (Lines 184, 205, 223).

Q16.  Line 101, They are trained or untrained? These 10 panelists are enough? 

Answer: Comments have been taken into account. They were all trained in professional food sensory quality. According to the following reference, 10 panelists were enough.

  1. 1. Siripatrawan, U.; Noipha, S. Active film from chitosan incorporating green tea extract for shelf life extension of pork sausages. Food Hydrocolloids2012, 27, 102-108, doi:10.1016/j.foodhyd.2011.08.011.

Q17. Give citations for the 2.7.1

Answer: Comments have been taken into account. It was supplemented accordingly (Line 115).

Q18. Provide the citation for section 2.7.2

Answer:Comments have been taken into account. It was supplemented accordingly (Line 123).

Q19. Line 123, which bag and how?

Answer: Comments have been taken into account. It was supplemented accordingly (Lines 131-135). The fish sausage was put into a retort pouch and then cooked in an electric thermostatic water bath.

Q20. Provide the citation for section 2.7.3

Answer: Comments have been taken into account. It was supplemented accordingly (Line 131).

Q21. Section 2.8 is how what is different from section 2.5.

Answer: Comments have been taken into account. The same machine was used for the determination of texture properties and gel strength, but with different probe, test mode and set-up parameters.

Q22.  Provide the citation for section 2.8

Answer: Comments have been taken into account. It was supplemented accordingly (Lines 145).

Q23. What is section 3 on page 5 that is given only results? usually, it should be Results and the discussion or discussion section should be there, but I can not find it. 

Answer: Comments have been taken into account. It was supplemented accordingly  (Line 178).

Q24. In section 3.1.1. given the sensory and gel strength together?

Answer: Comments have been taken into account. Surimi product quality, consumer acceptance and economic value are strongly influenced by its gel properties and flavor. Since the trend of sensory score and gel strength peaked at the same proportions of the three additives in the single-factor experiments, gel strength was used as the representative measurement index for the experimental design tests. They were supplemented accordingly(Lines 181-183, 247-249)

Q25. Line 190, Citation is given in lowercase

Answer: Comments have been taken into account. It was supplemented accordingly (Ref. 26).

Q26. Line 193, what is the sensory score? why not given? 

Answer: Comments have been taken into account. We have revised the manuscript as suggested (Line 207). The sensory evaluation score of fish sausage varied with the proportion of potato starch added, low scores at 4 % and 12 % (77.63 and 82.03, respectively).

Q27.  Discussion is very poor need to work on the discussion of the results.

Answer: Comments have been taken into account. They were supplemented accordingly (Lines 181-183, 219-221, 228-230, 266-268, 270-274, 281-285, 300-302, 320-326, 344-346, 357-359, 363-364, 369-370).

Q28.  conclusions should be fine-tuned.  

Answer:Comments have been taken into account. We have revised the manuscript as suggested (Lines 385-390).

Q29. The references should be revised properly. for instance check the 33 number the title of the paper is upper case.

Answer:Comments have been taken into account. We have checked all the references and formatted them according to the Guide for Authors (Ref. 3, 10, 14, 16-18, 22, 24, 28, 29, 31, 35, 39, 40).

To sum up, we have seriously considered your questions, and made modifications and explanations one by one, and the article has been improved, please do not hesitate to contact us if there are any question. Thanks again to the reviewers and editors for your hard work! Best wishes to you!

Address: Fujian Agriculture and Forestry University, College of Food Science, Fuzhou, Fujian, P. R. China 350002.

Tel.: +86 591 83736738

fax: +86 591 83739118

E-mail address: [email protected] (Y. Zhang), [email protected] (H. Zeng)

Sincerely,

Zhang Yi and Hongliang Zeng

17-Oct-2022 

Round 2

Reviewer 1 Report

The authors made some modifications. However, many issues were "planned to be published in the next paper" or "studied in the previous period". Therefore, I doubt the integrity of the current work. I strongly suggest the authors do a deeper discussion in the manuscript but not in the response letter.
